# Visuo-haptic processing of unfamiliar shapes: Comparing children and adults

**Furat AlAhmed[1], Anne Rau[2,3], Christian Wallraven[1]** *

**1** Department of Brain and Cognitive Engineering, Korea University, Seoul, Republic of Korea, **2** Department of Psychology, Eberhard Karls University of Tübingen, Tübingen, Germany, **3** Department of Psychiatry, University Hospital Tübingen, Tübingen, Germany

* wallraven@korea.ac.kr

**Data Availability Statement:** The OSF repository has been made public and can be found at: https://osf.io/g6t3s/?view_only=2131fadd15f54a3a9763a098a8f8e11c.

## Abstract

The question of how our sensory perception abilities develop has been an active area of research, establishing trajectories of development from infancy that last well into late childhood and even adolescence. In this context, several studies have established changes in sensory processing of vision and touch around the age of 8 to 9 years. In this experiment, we explored the visual and haptic perceptual development of elementary school children of ages 6–11 in similarity-rating tasks of unfamiliar objects and compared their performance to adults. The participants were presented with parametrically-defined objects to be explored haptically and visually in separate groups for both children and adults. Our results showed that the raw similarity ratings of the children had more variability compared to adults. A detailed multidimensional scaling analysis revealed that the reconstructed perceptual space of the adult haptic group was significantly closer to the parameter space compared to the children group, whereas both groups' visual perceptual space was similarly well reconstructed. Beyond this, however, we found no clear evidence for an age effect in either modality within the children group. These results suggest that haptic processing of unfamiliar, abstract shapes may continue to develop beyond the age of 11 years later into adolescence.

## Introduction

Sensory development starts very early and increases rapidly during infancy as for example witnessed through changes in haptic exploration for texture and weight information decoding as the hands are growing [1–3]. Similarly, in the visual domain [4, 5] suggested that infants as young as 6 months old have the ability to retain visual stimuli after a delay when presented as one of three options. Concurrently during development, visual acuity and contrast sensitivity improve until the ages of 5–6 years, whereas more explicit shape information is not taken into consideration during discrimination and feature extraction until the age of 7 years [6]. Interestingly, *multisensory* integration in this context is relatively slower compared to other processes during development [7, 8] showed that children can be as old as 12 years before using cue integration to reduce uncertainty in visual spaces, while before that children mostly rely on one single cue. This was also evident in an experiment in the haptic domain by [9] where in

**Funding:** This study was supported by the National Research Foundation of Korea under project BK21 FOUR and grants NRF-2022R1A2C2092118, NRF-2022R1H1A2092007, NRF-2019R1A2C2007612, as well as by Institute of Information & Communications Technology Planning & Evaluation (IITP) grants funded by the Korea government (No. 2017-0-00451, Development of BCI based Brain and Cognitive Computing Technology for Recognizing User's Intentions using Deep Learning; No. 2019-0-00079, Department of Artificial Intelligence, Korea University; No. 2021-0-02068, Artificial Intelligence Innovation Hub).

**Competing interests:** The authors have declared that no competing interests exist.

a haptic size conflict experiment, 6-year-olds relied completely on vision to answer, but 9- and 12-year-olds incorporated haptic cues to answer the questions. In [10] children categorized and grouped a family of alien "Beanbods"—there, it was shown that at the age of 6 years, children relied more on haptic cues than on visual cues for grouping. Other studies on multisensory visuo-haptic integration cite ages of 8–9 years as a point for the visual and haptic senses to match in performance, such as [11] in which children performed a visual and haptic comparative task using the Seguin Form Board, and [12] in which the children were presented with a height and orientation discrimination task of blocks. In a series of experiments [8] suggested that cue integration even within a single modality occurs around the age of 12 years based on a visual slant discrimination task.

While few studies have compared older adults to children in sensory processing, some have shown that as adults age, their sensory processing and sensitivity slow down, a decline that is especially visible for touch sensitivity [13]. To some degree this decline can be compensated, however, as for example witnessed in special participant groups such as surgeons, where constant training can maintain the level of sensitivity and accuracy [14]. In a study comparing children to adults in a multisensory, audiovisual task [15], showed that older children (11 years old) performed better than younger ones (8 years old) in both accuracy and speed of learning novel stimuli, with adults outperforming both groups of children, indicating continued development beyond 11 years. [16] compared adults' and children's Topological Property (TP) processing patterns in central and peripheral vision and found that at the age of 10 years and older their processing of patterns was as fast as the adults', while children between the ages of 6–8 years were slower and showed lower priority to TP elements in peripheral vision.

As evident from the previous discussion, the elementary school age range seems to be a pivotal time point in the development of sensory processing skills—in the present study, we therefore focused our investigation on the development of haptic and visual evaluation skills, comparing both elementary school children and an adult control population. For this, we chose a similarity rating task that involved processing of unfamiliar shapes either in vision or touch. With this task, several studies in adult populations have shown that both tactile and visual exploration of complex shapes result in similarity rating patterns that are remarkably similar across modalities and that are even capable of capturing an underlying parametric structure that was used to generate the shape information [17–21]. Despite this evidence, however, comparatively little is known about the *developmental trajectory* of such shape processing in children, which is the main purpose of the present study.

## Materials and methods

### Participants

Participants were recruited at one elementary school in Tübingen, Germany, in collaboration with the Developmental Psychology Lab at the University of Tübingen through voluntary participation Parents of the participants were given a description of the experiment (which they were instructed not to share with the child) together with a consent form. A total of 125 children participated in this experiment (72 women, 53 men) between the ages of 6.0 and 10.5 years old (average age = 8.4 years, standard deviation = 1.2 years). All participants were either native German speakers or had a good understanding of the language, had no obvious neurological deficit, and had normal or corrected-to-normal vision. The design followed a between-participant approach with half of the children randomly allocated to a visual group (63 children) and the other to a haptic group (62 children). Three participants failed to complete the task and were excluded. The data from a further two participants in the visual group was excluded as they did not follow the instructions correctly, yielding a total of 60 children in

both groups. All children received a small toy as compensation for their participation after the experiment.

Adult participants were recruited through an online post and were all Korea University undergraduate students (13 women, 17 men, average age = 22.8 years, standard deviation = 2.2 years, range between 20.2 and 24.5 years) and the experiment received approval from the Korea University Institutional Review Board (IRB number: KU IRB-2019-0107-01). A total of 15 participants performed the haptic task and another 15 participants performed the visual task. All participants were healthy, had no obvious neurological deficit, and had normal or corrected-to-normal vision. Written informed consent was obtained after the briefing from experimenters before the experiment was conducted.

The sample size for the adult group was set similar to that of studies using this well-documented paradigms with adult participants [17–21]—the sample size for the children group was set to be larger given the potentially larger variability in performance and different attention span following previously published research (e.g., [10, 12]).

## Stimuli

The stimuli used in this experiment were generated by a variation of the "superformula" model to create a two-dimensional cross within a three-dimensional cube of a parameter space. The shapes were created by varying 3 parameters in 2 different dimensions ($n_1$, $m_1$, and $m_2$ in the inset equation (1)) to create 9 different shapes (shown in Fig 1 in their position in parameter space). We note that the parameters introduce global shape changes that are hard to summarize easily—for more discussion on these stimuli, see [21]. The average measurements were 7.59 × 8.56 × 7.56 cm with an average weight of 200 g. The stimuli were printed as polyamide objects on a 3D printer (Zprinter650, 3DSystems, USA) for use in both visual and haptic

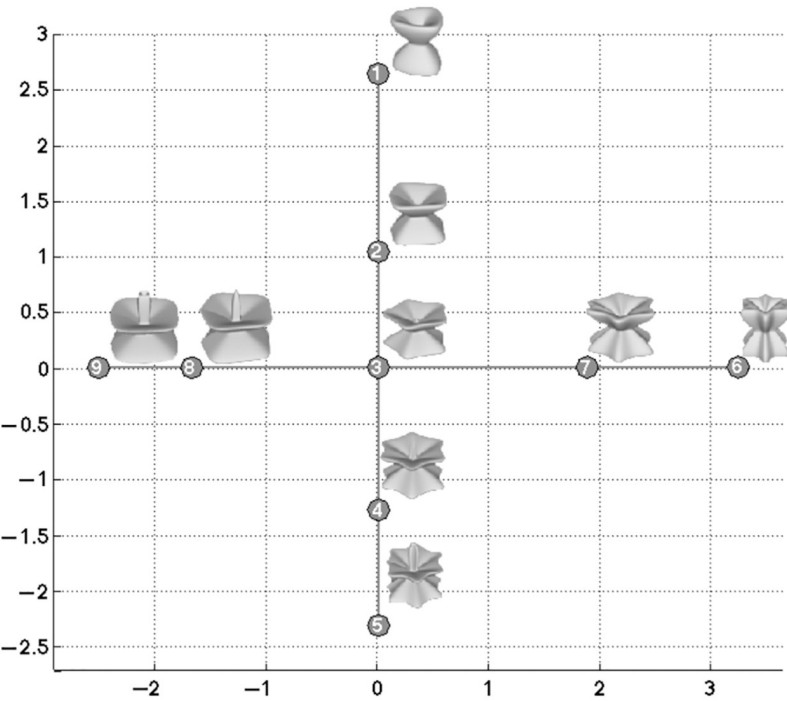

**Fig 1. The distribution of the 9 selected shapes for the experiment.**

experiments.

$$r(\varphi) = \left[ \left| \frac{cos\,cos\left(\frac{m\varphi}{4}\right)}{a} \right|^{n_2} + \left| \frac{sin\,sin\left(\frac{m\varphi}{4}\right)}{b} \right|^{n_3} \right]^{-\frac{1}{n_1}}$$

$$x = r_1(\theta)cos\,cos\,(\theta)\,r_2(\emptyset)\,cos(\emptyset)$$

$$y = r_1(\theta)sin\,sin\,(\theta)\,r_2(\emptyset)\,cos(\emptyset)$$

$$z = r_2(\emptyset)\,sin(\emptyset)$$

## Setup

**Children similarity task.** The experiment for the children group was conducted in a designated empty room in the school during school hours. The children were shown into the room and the overall experimental schedule was verbally explained by the experimenters.

The first part of the haptic experiment consisted of a quick measurement of the two-point discrimination threshold by using a caliper and pressing it lightly against the index finger of the dominant hand of the child. The distance between the points started at zero and incremented until the child was able to report feeling 2 points pressing against their index finger.

After this, they were also made familiar with the rating scale that was written on a paper in German that was to be used for their responses; the scale was a 5-point scale, where the numbers corresponded to 1 = "very dissimilar", 2 = "dissimilar", 3 = "neutral", 4 = "similar" and 5 = "exactly the same". A 5-point scale was selected to limit the options and to make it easier for children to decide on ratings.

Before the rating task started, participants were first presented with all nine shapes once in random order to fully explore via touching (Haptic group) or to fully explore via looking (Visual group). Other than trying to fully explore the stimuli, participants were not given any explicit instructions as to how the shapes should be explored in either condition.

*Haptic task*. A fixture on a desk with a curtain was used during the haptic exploration through which the participants were asked to extend their dominant hand for exploration. During the main task, the objects were placed on the table at a comfortable handling distance of about 0.4m away (Fig 2) where the participants explored them haptically for 6–8 seconds after which the first object was replaced by the second object for comparison. The rating was given verbally after two objects were presented and recorded by the experimenter.

*Visual task*. During the visual experiment, the objects were placed in front of the participants at the same distance, and they were instructed to explore them visually only. Participants were allowed to move their head a little but otherwise were instructed to not get up from the chair during visual exploration. The exploration time for each object was kept similar to the haptic condition, and objects were kept behind a barrier to prevent participants from seeing them during the experiment. Ratings were again given verbally after each object pair and recorded by the experimenter.

For both conditions, it was decided that it was better to have the experiment timed by the experimenter and not by a computer program to remove the stressful time factor from influencing the children's responses. The full experiment consisted of 45 trials resulting from all pairwise comparisons of the nine objects, including comparisons of objects with themselves ("same trials"). The total time for the experiment was between 20–30 minutes depending on how many breaks a child had taken. Children were encouraged by the experimenters to take breaks whenever they needed them.

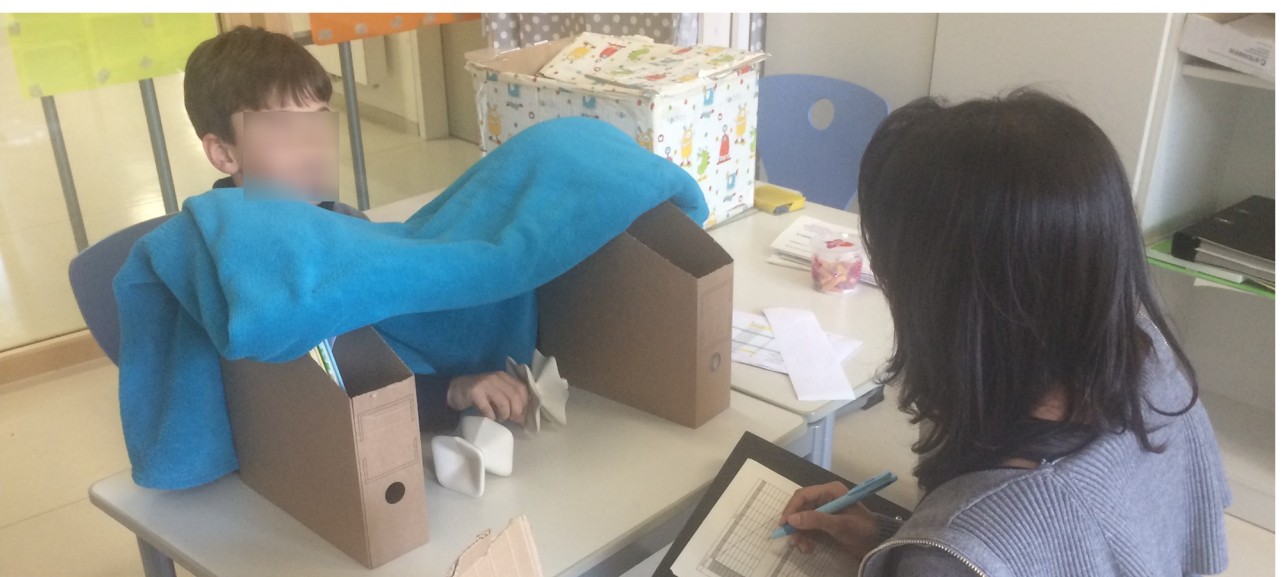

**Fig 2. Setup for the haptic condition.**

After the experiment ended the children were asked about the difficulty of the task and almost all answered that it was easy. When asked if they had a strategy in rating the objects, the answers varied considerably, with some children mentioning to count the "folds" and trying to recognize the shape of the "stem". Given the time constraints placed on this experiment, it was not possible to conduct a more detailed, structured post-experiment debriefing unfortunately.

**Adults similarity task.** The similarity rating task for the adults was almost identical to the children's setup with the exception of adult participants performing two repetitions of the task (here, we analyze and report only the first repetition), and the use of a numeric 1–5 scale instead of a verbal scale. The adults' experiment was run by a computer program that recorded the trials. Here, each trial lasted 12 seconds (6 seconds per object) with an additional 3 seconds for answering. The total time for the experiment including breaks was 25 minutes.

## Analysis

The individual ratings of each participant for 45 trials (all trials for the children and only the first repetition for the adults) were analyzed in the following. Ratings were first correlated across all participants via Spearman correlations for all pair-wise comparisons within each modality group to check for inter-rater consistency. We next also correlated the responses for the 45 comparisons across participant groups (but within modality) for all pairwise comparisons to gauge how well response patterns would match. Bootstrapping was done to determine confidence intervals for these comparisons. Since correlations only look at relative changes in responses, we next conducted standard t-tests comparing the adults and children for each of the cells of the 9x9 (symmetric) similarity matrix for each condition, followed by correction for multiple comparisons.

An analysis of the same-object comparisons was conducted next to determine a ground-truth "accuracy" response as the similarity rating for these trials should be "5". Two-sample t-tests were run to compare average same-trial ratings across modalities and groups. Linear regressions were then conducted for the children participants to see whether there would be any trend in accuracy as a function of age.

The next set of analyses focused on the similarity matrices and the resulting perceptual space. First, the similarity matrices were compared to those obtained from the actual physical parameters used during stimulus creation to check how well they would match. Next, multidimensional scaling (MDS, e.g., [17]) was applied to investigate the topologies of the resultant perceptual spaces. Stress values of the MDS and bootstrapped fits of the spaces against each other using Procrustes analysis were performed to determine the degree of topological similarity.

All data were analyzed using MATLAB (R2019a, The Mathworks, Natick, MA, USA).

## Results

### Response consistency within modalities

Consistency across participants was the first step in analysis to assure the reliability of the results. To measure participants' rating consistency, their similarity rating responses for all object pairs were correlated to each other resulting in the correlation matrices shown in (Fig 3). By averaging their lower diagonal, response consistency yielded an average correlation of $r$ = .56 for haptic (Fig 3a), and $r$ = .55 for visual exploration (Fig 3b). In Fig 3, all results are sorted from the youngest to the oldest age to potentially highlight differences—any consistent change across age (such as improvement in consistency for older children) would be visible as a pattern in the correlation matrix colors—this, however, was not observed.

The adults' responses were more consistent compared to children with averaged correlations of .734 for haptic (Fig 4a) and .749 for visual exploration (Fig 4b).

### Response consistency across participant groups

Rather than looking at consistency within groups, one can also look at the consistency across groups, where the comparison of children to adults is of particular interest for assessing potential differences across developmental stages. A bootstrapped correlation analysis for the two haptic groups yielded $r$ = .60 (confidence interval [.36, .75]), and $r$ = .55 (confidence interval [.29, .72]) for the two visual groups, respectively. The results of the bootstrap distributions of the correlations are shown in Fig 5. As the two confidence intervals overlap, the relative rating pattern between children and adults was similar across modalities.

Correlation results only pay attention to the relative differences in the rating profiles, but do not assess absolute differences in the rating behavior. To address this in more detail, t-tests

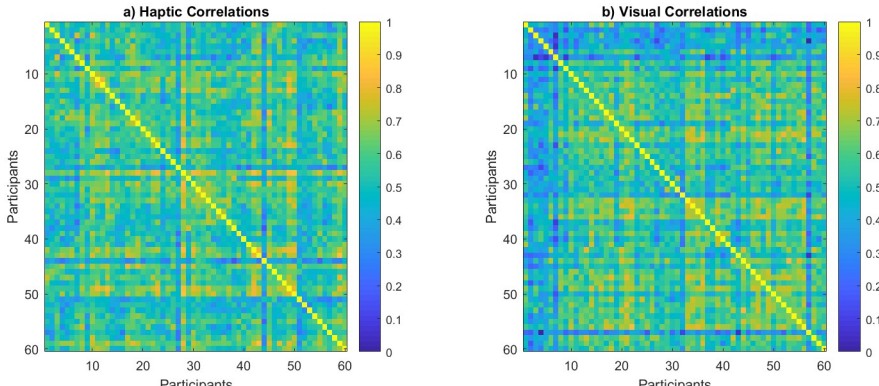

**Fig 3. Correlation across child participants, sorted from youngest to oldest.** (a) Haptic group. (b) Visual group.

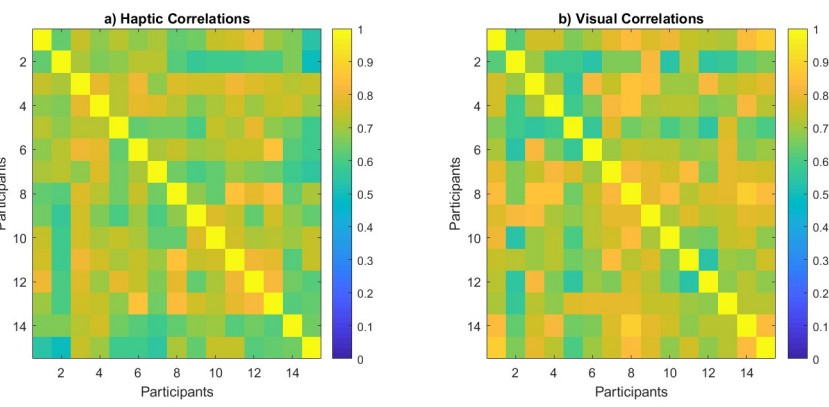

**Fig 4. Correlation across adult participants.** (a) Haptic group. (b) Visual group.

were run comparing each of the 45 cells of the similarity matrix across modalities of the two groups.

Given that there were 45 trials to be tested, a Benjamini-Hochberg FDR correction for multiple tests was applied. For better visualization of the effect sizes of potential ratings differences, we also subtracted the children's averaged ratings from adults' averaged ratings for each of the 45 entries. Fig 6 shows these effect sizes visualized in the 9x9 similarity matrix format with colored cells being those that survived the FDR correction. More differences in absolute ratings were found for the haptic modality—in addition, rating differences were observed for different sets of object comparisons in both modalities. In both cases, however, the main effect was that children's ratings were higher, that is, they rated the objects as more similar than adults.

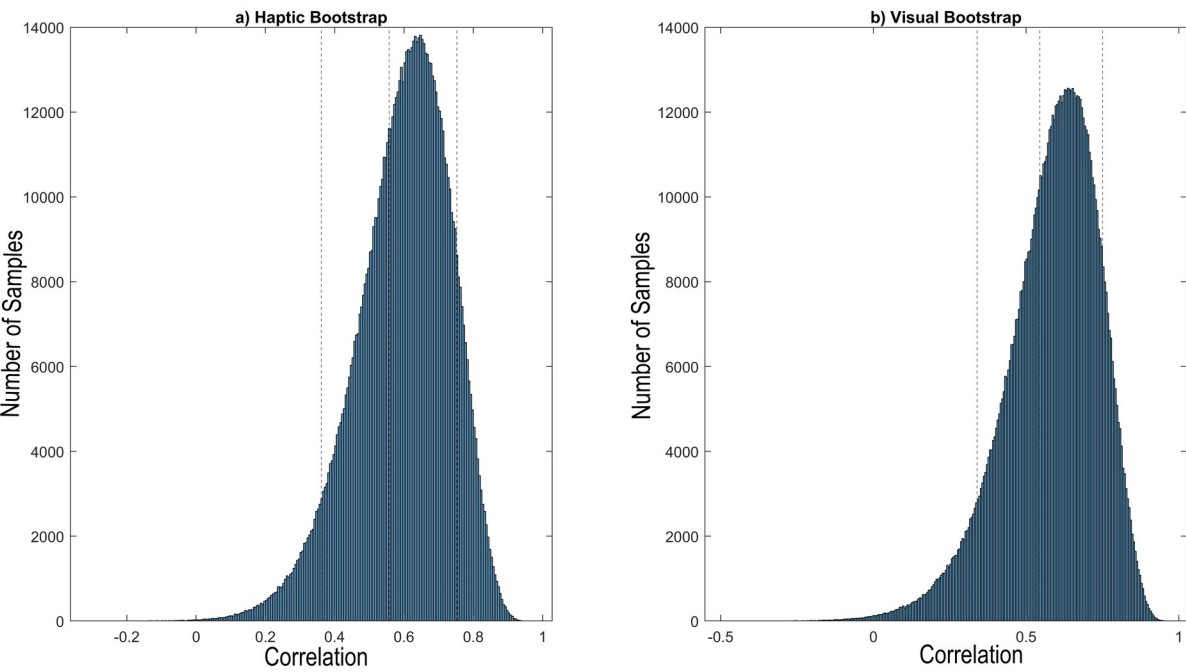

**Fig 5. Bootstrapped correlations for a) haptic and b) visual groups.**

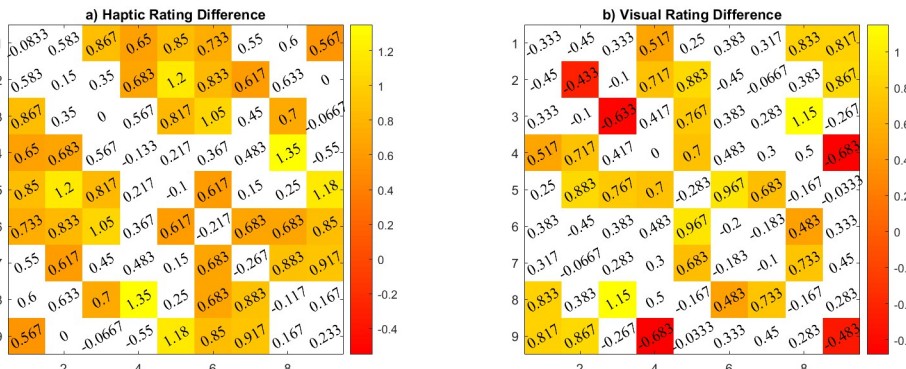

**Fig 6. Difference between averaged children and adult ratings.** Cells in color are the cells that survived the FDR correction, the value inside is the observed rating difference between children and adults' ratings. The white part of the color bar indicates non-significant rating difference ranges according to FDR.

It is also possible that the children formed a strategy of counting "petals" and comparing the "stems" and proceeded with it throughout the experiment while possibly ignoring other features such as size. Structured responses explaining how the children rated the similarity of the objects were not able to be collected, however, due to the time constraints set by the school.

## Same trial analysis

Trials in which the same object was presented twice ("same trials") represent a special case in our experiment, as they have a "true", expected response—participants should respond with the highest similarity rating in this case.

The average rating for these same trials for children was 4.64 (haptic) and 4.49 (visual). The difference in performance for the two modalities was moderately significant ($t(118) = -2.61$, $p = .01$, BF = 4.03), with a higher performance in the haptic domain.

The analysis for adults of the same trials yielded 4.72 (haptic) and 4.8 (visual). This difference in performance for the two modalities was not significant ($t(28) = 0.96$, $p = .35$, BF = 2.1).

When comparing across children and adults, same trial performance was better for the visual task in adults ($t(73) = -3.14$, $p = .002$, BF = 31), but similar for the haptic task ($t(73) = -1.05$, $p = .30$, BF = 2.22).

In a more detailed analysis looking for potential age effects in the children group, a linear fit was calculated for both haptic and visual conditions. Fitting the responses of all the participants against the ages in months, we found no evidence of significant differences or improvements across age in the haptic group ($r = -.05$, $p = .70$), but there was a borderline significant improvement for the visual group with age ($r = .25$, $p = .05$—see Fig 7).

## Analysis of age effects in children group in all trials

Following the previous finding that indicated some effects of age for the same trials, we next ran regressions for each of the 45 entries in the similarity matrix in each of the modality conditions for the children group. The resulting p-values of the fits, however, did not show any significant change in the children group beyond those reported above.

Overall, these results show that age-related changes in the similarity ratings were at best small and potentially limited to a better discriminability of the same stimuli (see previous analysis of averaged same trial data).

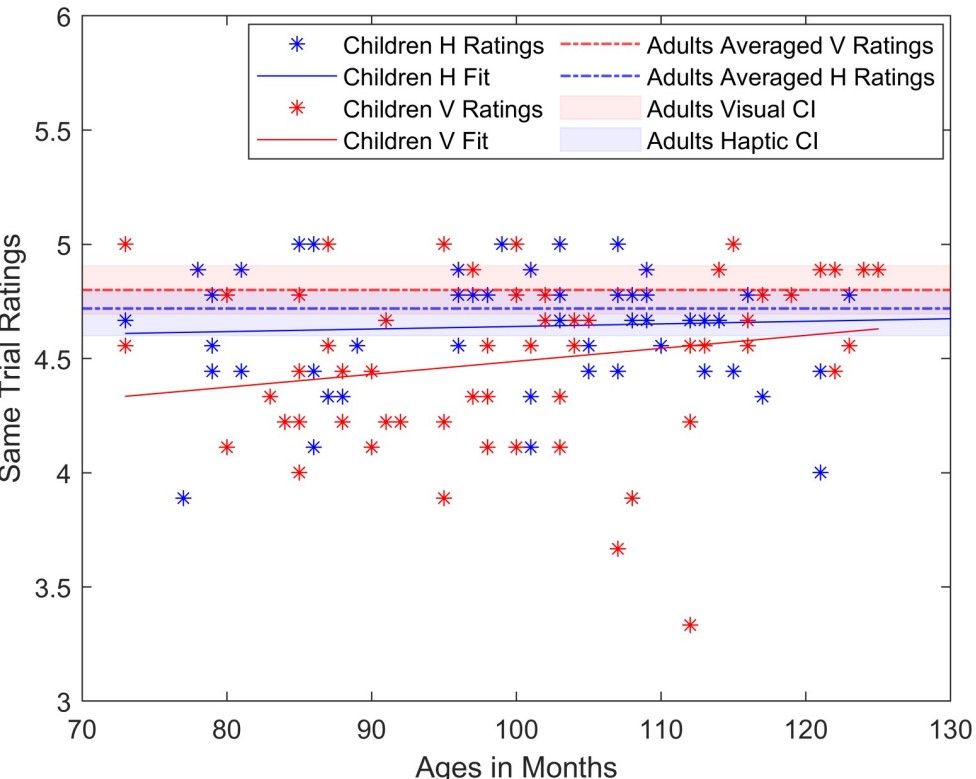

**Fig 7. Same trial ratings plotted for all groups (age in months apply to children only).** Dashed lines represent the adults averaged ratings.

## Analysis of age effects related to two-point discrimination

To search for another possible point of perceptual improvement during early childhood, the 2-point discrimination test was looked into. Linear fitting showed no significant trend (Fig 8) in our sample and age range ($r = -.03$, $p = .84$), results that are in line with [22].

There also seemed to be no significant relationship between the 2PD test and performance on haptic same trials ($r = -.09$, $p = .50$).

## Analysis of perceptual spaces

To compare participants' similarity perception with the underlying parameter space, averaged similarity matrices in the two modality conditions (Fig 9a and 9b) were correlated to the similarity matrix derived from the Euclidean distance between the parameter values of the nine objects (Fig 9c). The resulting correlations were high in both cases with r = .77 (haptic) and r = .84 (visual).

Averaged adults' raw responses correlated to the physical space (Fig 10) resulted in $r = .79$ for haptic (Fig 10a) and $r = .79$ for visual exploration (Fig 10b).

These results show that both children and adults were capable of estimating the distance between the objects in both modalities to a high degree, which at first glance is compatible with earlier work [17–21].

**Multidimensional scaling.** To compare the topologies of the perceptual representations in more detail, nonclassical multidimensional scaling (MDS) was next used [17] on

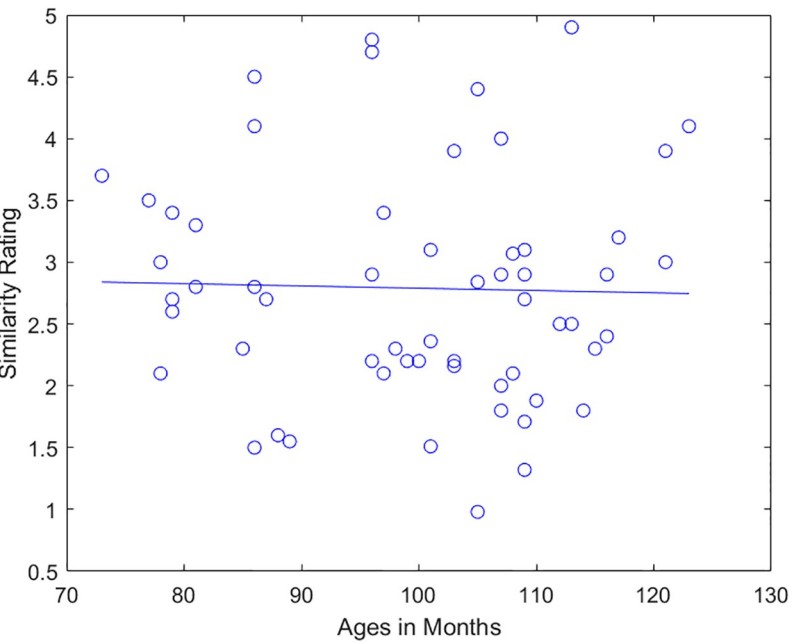

**Fig 8. Two-point discrimination polynomial fit.**

condition-averaged similarity matrices. The stress values obtained for judging the goodness-of-fit of a two-dimensional scaling solution for the children data were s = 0.04 for the haptic condition and s = 0.04 for the visual condition which is determined as a "good" fit [17, 23].

The resultant perceptual space was fitted to the solution space obtained from the parameter values via a Procrustes transformation (see [18] for a similar method). The resultant Procrustes value "d" can also be used to assess the similarity between the spaces, where smaller values indicate better congruence. As before, fitting was done in bootstrapped fashion to yield confidence interval estimates. Fig 11 shows the resultant average haptic perceptual space of the children's responses in blue and the visual space in red. The Procrustes d

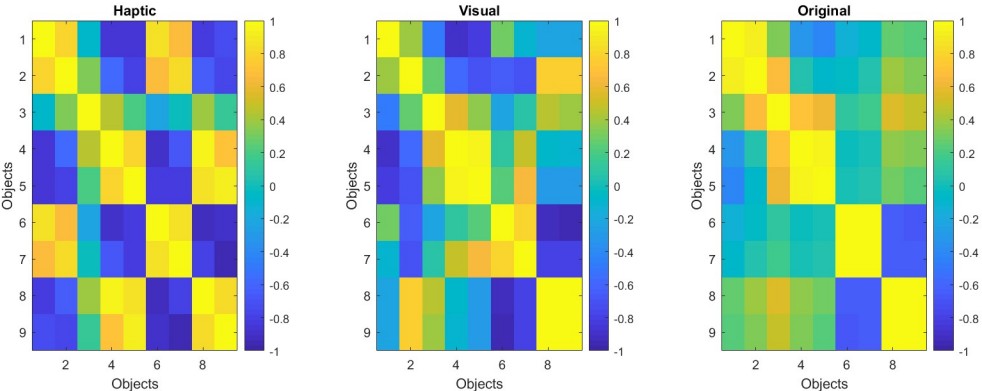

**Fig 9.** Averaged children's responses correlated (a) haptically and (b) visually. (c) shows the original distance.

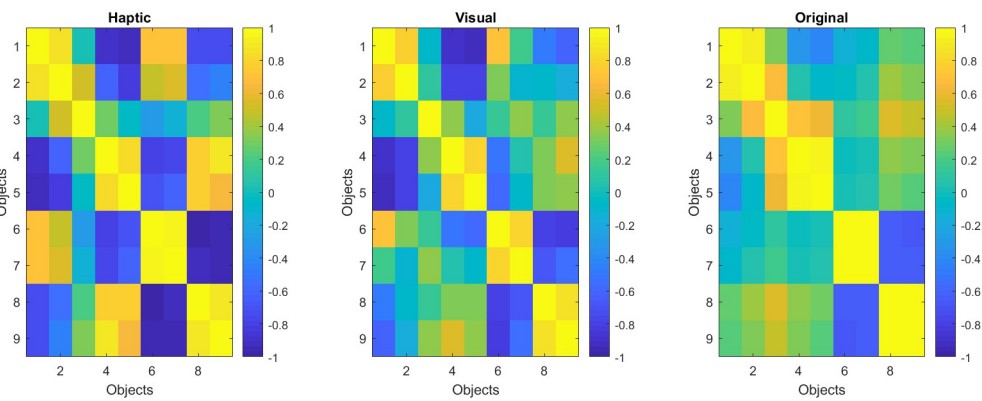

**Fig 10.** Averaged adult responses correlated (a) haptically and (b) visually. (c) shows the original distance.

comparing these spaces to the underlying gray parameter space were haptic: d = 0.38 and visual: d = 0.11.

Applying the same analysis procedure to the adults' dissimilarity matrices, we obtained stress values of s = 0.08 for the haptic condition and 0.08 for the visual condition—again signaling a "good" fit. The resulting haptic scaling component was d = 0.27, and the visual scaling component was d = 0.14 (Fig 12).

To compare MDS results further across modalities, Procrustes' fitting was applied in a bootstrapped fashion for children and adult data. This was done by repeating the analysis 1000 times, each time leaving out half of the participants for the calculation of MDS spaces and aggregating the results over repetitions. This procedure yielded similar averages as above: d = 0.40 and d = 0.12 for the children's haptic and visual results respectively, and d = 0.24 and d = 0.13 for adults.

The boxplots of the children's results after this bootstrapping procedure in Fig 13 show that children's fit quality to the parameter space was inferior to that of the adult group. Similar, and

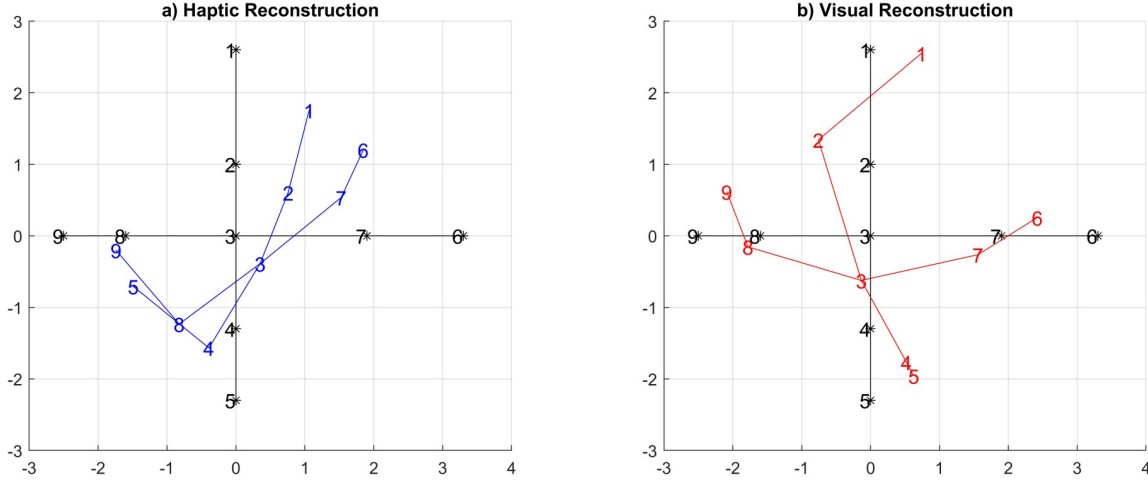

**Fig 11. Children resultant MDS spaces.** a) Haptic distribution b) visual distribution.

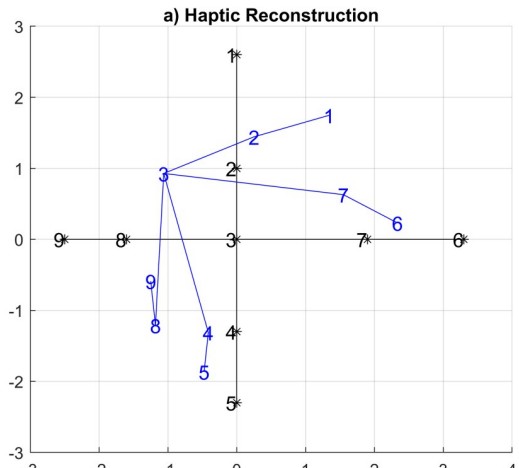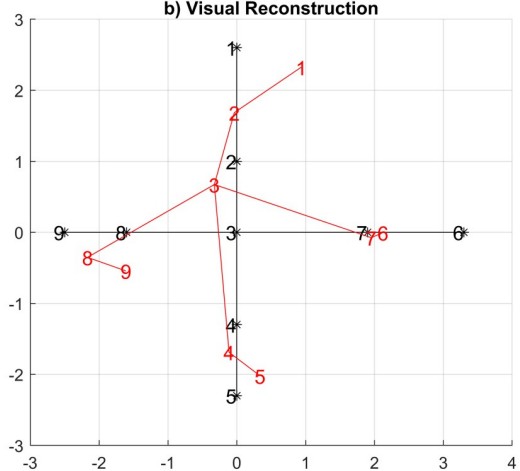

**Fig 12. Adults' resultant MDS spaces.** a) Haptic distribution b) visual distribution.

overall better, fit qualities, however, were obtained for the visual modality in both groups. These results suggest that the children's ability to reconstruct perceptual spaces cognitively is comparable to that of adults in the visual condition, however, it seems to be inferior to that of adults in the haptic condition.

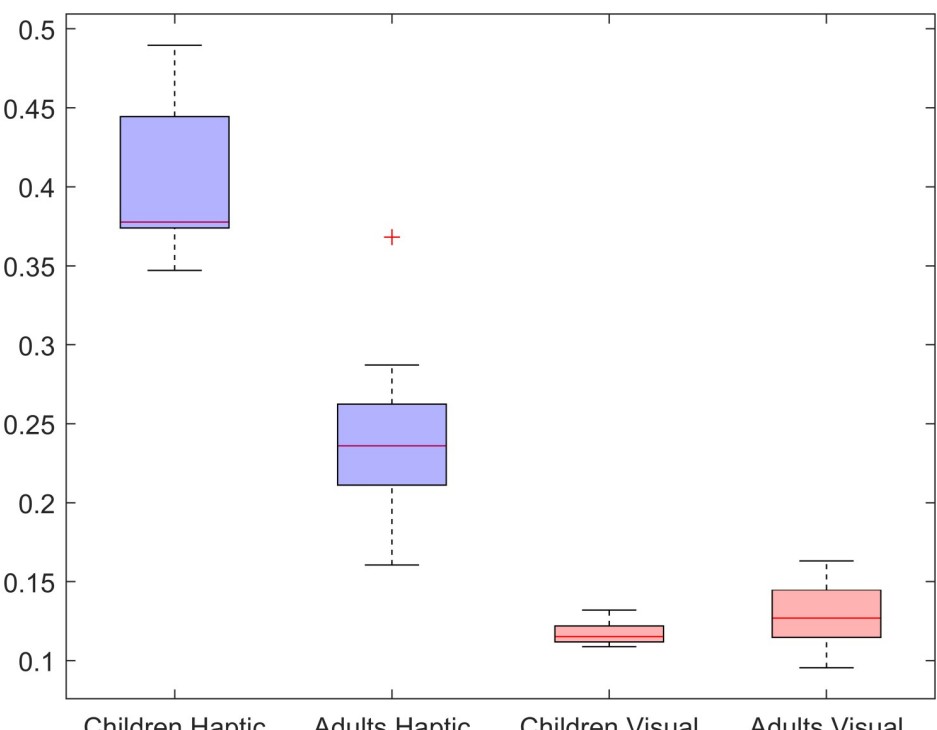

**Fig 13. Boxplots bootstrapped MDS data showing the error in fit.** The "+" is an outlier data point according to the standard box-whisker plotting parameters.

## Discussion

The present study investigated developmental aspects of haptic and visual shape processing, within the age range of elementary school children and with an adult control group as baseline. Both groups were presented with unfamiliar shapes that were parametrically distributed in a physical space, and pairwise ratings were obtained from both groups to examine the differences.

Both groups' responses were consistent within their own group and across the other, suggesting that participants rated the distance between the objects in a largely similar manner. Nonetheless, overall adults had a more consistent range of rating values. Comparing individual ratings for detailed differences, the children's similarity ratings were higher in absolute terms compared to the adults as indicated in Fig 6. These findings are similar to those of [24] where a group/family of unfamiliar objects (Widgets) that had a structure with eyes, crest, tail and legs attached were categorized visually and haptically by groups of children, adolescents and adults. The children's performance was as good as the adults' and there was no advantage of age in the categorization task.

There were only minor differences in accuracy as measured for the "same trials" between the modalities in our experiment. On average, adults showed a slightly better performance in the visual trials compared to the children population, but there was no difference in accuracy for the haptic modality. This is similar to the results of [25] where children were able to detect incongruences in faces in a holistic manner just as well as adults, and [26] in which the ability to differentiate the duration of visual and auditory signals was comparable in adults and children as young as 5.

One of the main findings of our experiment was a clear difference between adults and children in terms of the reconstructed perceptual spaces. Here, we observed that the *visual* perceptual reconstructions were highly similar between children and adults, indicating little further development beyond the age of 6 even. In contrast, there were clear qualitative differences in the *haptic* perceptual spaces between the children and adult groups. From the MDS results in Fig 11 and a qualitative analysis of potential stimulus properties, it seems that the children may have relied more on fast-outstanding features than on less outstanding details. Returning also to the results of Fig 6, we observe that in the haptic reconstruction they placed object #3 closer to #7 and #4 (this could probably be due to the petal-like edges) and #8 closer to #4 (this could be due to the type of the "stem" on both objects). Both of those features were recurrent criteria mentioned by children spontaneously after they provided feedback about the experiment. A more in-depth analysis of the introspective features of children versus adults in this task, however, was not possible in the context of the present study, given the time constraints in running the experiment in the school and hence is left for future work. We do note in addition that changes in feature processing were, for example, also observed in [10].

To look further for a possible transitional point, age group differences among children were also explored. In this experiment the age ranged from 6 years and 1 month to 10 years and 9 months, and only a slightly significant improvement was detected in the visual modality for same-trials, but none for any other setting. This may be in line with [27, 28] where the visual performance slowly improved in this age range in an object recognition task for both healthy and PVL (bilateral Periventricular Leukomalacia) diagnosed children. Notably, however, in their experiments, performance in both modalities increased faster at around the age of 8–10 years, reaching adult levels—an effect we do not observe in our data, as for example, the visual perceptual space is highly similar between children and adult groups. A crucial difference between their experiment and ours, however, is that the objects for our

study are unfamiliar, abstract objects, whereas in [27, 28] stationary and household items were used. Hence, in [27, 28] the objects would have been even more familiar after the children had been in school for a few years as a result of lessons and further social interaction with adults. In contrast, the objects in our experiment were not familiar and therefore may have required a more abstract thinking ability that may possibly develop in later childhood and adolescence.

In this context, a large difference between 7–8 and 9–10 age groups was found in a semantic fluency experiment by [29] that included children as old as 16. Children show verbal improvement around 3–6 years old, like that in the results of [30] where the older children were asked to list as many animals, foods and musical instruments in 60 seconds. In [31] groups of 8, 12 and 21-year-olds were asked to list as many animals and furniture items as they could in 7 minutes, and the results showed that while the adults were capable of listing more items, the categories and semantic cluster retrieval processes were similar to the younger children group. This research could also explain why the children group in this experiment did not exhibit changes around a certain age as they did in other experiments such as [27, 28] with continuously familiarized objects—this is because verbal fluency development would have an impact for a familiar object task, whereas the objects in our experiment were unfamiliar making it potentially more difficult to attach clear semantic associations.

One potential limitation of our experiment is that we were only able to gather results from a single trial per comparison due to time restrictions and concentration of the younger children. In this context, it is interesting to note that the second trial for adults did not show differences in comparison to the first trial (correlation of r = .87 of first repetition to second repetition in the ratings). Similarly, the children's same trial ratings were consistent at similar levels to those of the adults, showing that the ratings per se were not unreliable.

Another possible issue is that the reliability of the children's similarity ratings may be limited given the potential complexity of the concept of "similarity"—indeed, we did observe a greater variability in ratings for children. Nonetheless, all children in our experiments seemed to grasp the concept of the task fairly quickly and found the experiment overall not difficult to complete. In this context, studies from the developmental literature show conflicting results: on the one hand, 5-year old children in [32] had higher sensitivity for detecting differences in recorded speech using a paired-comparisons method. On the other hand, in [33] a similarly-aged group of children were not able to generate typicality judgements even though the task was introduced as a simplified game, and suggested that the family resemblance method is better for obtaining answers from children. Taking both findings into consideration, it was deemed more suitable to use a similarity task using a rating scale in a paired-comparison task because the aim of this study was to explore the abstract perception abilities instead of a categorization difference. Similarly, because our objects were not grouped into certain families, but were actually designed to form an abstract two-dimensional space, a grouped or a categorization task would not have been appropriate for the reconstruction of the space.

To summarize, in this experiment we looked into the developmental trajectory of haptic and visual processing for unfamiliar objects as opposed to the integration of both senses in elementary school children. What we found is that, even though there may be a point of visuo-haptic integration between the ages of 8–10, separately there was no evidence of such a change in our data on unfamiliar shapes in kids as young as 6 years old. We also found evidence for a slight yet significant increase in performance across age for averaged "same trials" in the visual condition. It is evident from the MDS, however, that the children's perception does not match

the adults' in the haptic modality, suggesting an older point of change in perception in unfamiliar shapes.

## Acknowledgments

The experiments were made possible thanks to the collaboration of the Developmental Psychology Lab at the University of Tübingen and the active cooperation of the Grundschule Innenstadt. We would like to thank all children for their participation, all teachers for their willingness to break class for the participants, and especially the headmaster of the school Hans-Martin Widmann for his support of our experiment.

## Author Contributions

**Conceptualization:** Christian Wallraven.

**Data curation:** Furat AlAhmed, Anne Rau, Christian Wallraven.

**Formal analysis:** Furat AlAhmed, Christian Wallraven.

**Funding acquisition:** Christian Wallraven.

**Investigation:** Christian Wallraven.

**Methodology:** Furat AlAhmed, Anne Rau, Christian Wallraven.

**Project administration:** Anne Rau, Christian Wallraven.

**Resources:** Christian Wallraven.

**Supervision:** Christian Wallraven.

**Visualization:** Christian Wallraven.

**Writing – original draft:** Furat AlAhmed, Christian Wallraven.

**Writing – review & editing:** Furat AlAhmed, Anne Rau, Christian Wallraven.

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
