## [Decision Letter · Decision Letter 0]

9 Sep 2022

PONE-D-22-15932Visuo-haptic processing of unfamiliar shapes: comparing children and adultsPLOS ONE

Dear Dr. Wallraven,

Thank you for submitting your manuscript to PLOS ONE. After careful consideration, we feel that it has merit but does not fully meet PLOS ONE’s publication criteria as it currently stands. Therefore, we invite you to submit a revised version of the manuscript that addresses the points raised during the review process.

We look forward to receiving your revised manuscript.

Kind regards,

Mariella Pazzaglia

Academic Editor

PLOS ONE

Journal Requirements:

2. Please change "female” or "male" to "woman” or "man", or "girls" or "boys" as appropriate, when used as a noun (see for instance https://apastyle.apa.org/style-grammar-guidelines/bias-free-language/gender).

“This work was supported by Institute of Information & Communications Technology Planning & Evaluation (IITP) grants funded by the Korean government (MSIT) (No. 2019-0-00079, Department of Artificial Intelligence, Korea University; No.2021-0-02068-001, Artificial Intelligence Innovation Hub) and from the National Research Foundation of Korea (NRF; NRF-2017M3C7A1041824, NRF-2019R1A2C2007612).”

“This work was supported by Institute of Information & Communications Technology Planning & Evaluation (IITP) grants funded by the Korean government (MSIT) (No. 2019-0-00079, Department of Artificial Intelligence, Korea University; No.2021-0-02068-001, Artificial Intelligence Innovation Hub) and from the National Research Foundation of Korea (NRF; NRF-2017M3C7A1041824, NRF-2019R1A2C2007612). Grants were awarded to CW.

The funders played no role in the study design, data collection and analysis, decision to publish, or preparation of the manuscript.”

Additional Editor Comments:

As usual, I have invited comments from experts from your research domain.

As you will see, multiple limitations were highlighted by reviewers that would need to be addressed/explained carefully.

Taken altogether, let me invite you to prepare a revision that addresses the issues, together with a cover letter explaining how you did so. My plan is to resend the revision to the present referees.

Reviewers' comments:

Reviewer's Responses to Questions

**Comments to the Author**

1. Is the manuscript technically sound, and do the data support the conclusions?

Reviewer #1: Yes

Reviewer #2: Partly

2. Has the statistical analysis been performed appropriately and rigorously? 

Reviewer #1: Yes

Reviewer #2: No

3. Have the authors made all data underlying the findings in their manuscript fully available?

Reviewer #1: Yes

Reviewer #2: No

4. Is the manuscript presented in an intelligible fashion and written in standard English?

Reviewer #1: Yes

Reviewer #2: Yes

5. Review Comments to the Author

Reviewer #1: The article is original and well-written. The methodology is well done and the statistical analyses that have been performed are also adequate. the authors have also brought the most critical points into the discussion.

The only weak point of the article in my opinion is the small sample of adult subjects compared to children. Is it not possible to increase it? The results would be even more interesting

Minor revision:

page 10: The sentence "correlation results....." is not complete

Reviewer #2: Comments for the above questions:

#1. The authors present an interesting and well-designed study, in which they compared the similarity ratings of visually-presented and haptically-presented novel objects in children and adults. The authors observed that the similarity maps of haptically-perceived objects was different from the source objects and less reliable in children compared to adults– these differences were not observed in visually-perceived objects.

For this criteria there are two main issues.

(1) No justification was provided for the sample sizes (60 per group for children and 15 per group for adults)

(2) The authors’ concluding paragraph focuses on multisensory integration, and specifically “a point of visuo-haptic integration” – this is inappropriate considering integration was not investigated here. There is very little in terms of conclusions based on the numerous analysis presented in the paper.

#2. (Statistical analyses) Generally, there is information missing from statistical descriptions.

• There is no description of how the data was treated: each child participant provided 45 data points (45 comparisons) and each adult participant provided 90 data points (45 comparisons, twice). The authors only mention that ‘their responses were correlated to each other’ – were each individual ratings for each participant correlated to each individual rating for other participants? Was averaging performed across pairs that were more/less similar? Across each individual’s responses? Also, what type of correlation was calculated?

• It is not clear if only first trial or both trials are included in the analysis of adults’ performance. If both trials are included: it is possible that this stabilized participants’ responses and led to the finding that responses were more reliable in adults. If both adult responses were included, it would be important to report the analysis of only the first response, as it may change the results and the authors’ interpretation / conclusion.

• On page 8 the authors report that age differences were ‘not visible in the data’ – what statistical test was performed to support the statement?

• On most figures the axes are not labeled.

• On page 4, figures 3 and 4 occlude text. This also occurs on page 10: Figure 6 occludes text and overlaps with Figure 5, occluding it as well.

• On page 11-12, the authors perform two separate analyses on the averaged ratings for ‘same’ trials (visual vs. haptic in each age group, and then children vs. adults for each modality). The authors should select one set of comparisons.

• On page 12 it is not clear what data is being used to ‘fit the response of all participants against age in months’.

• Through the results section, the authors seem to begin interpreting their data (for example p 13 for age-related changes, p15 for estimated distances)

• On p. 14 it’s not clear whose data (adults, children, all) is being presented in the analysis of perceptual space. The same problem arises in the first paragraph for multidimensional scaling (page 16).

• It’s often not clear why specific analyses are being performed, for example, on p 17 the authors say “To compare MDS results further across modalities…” They then indicate they left out participants for the calculation, but did not explain which participants or how they were selected for exclusion.

• There is no report of how common specific children’s responses were about how they rated objects – though these are addressed in the discussion. It would therefore be important to describe these in the results

#3. (data availability) The authors report that ‘some restrictions will apply’ but do not describe these restrictions. The authors also indicate that data will be made available upon publication via a Google share link.

#4. Generally the quality of the English language is appropriate. However, the constant use of numbers in the middle of sentences to refer to specific authors makes reading the manuscript awkward.

On page 4, the authors should add ‘participants’ after the second 15, and ‘the visual’ should be replaced by ‘the visual task’ in the sentence “A total of 15 participants performed the haptic task and 15 (participants) performed the visual (task).”

Further to the above, the authors should address the following:

Introduction: The introduction is very short (2 pages / 3 paragraphs), quite superficial, and lacks focus. The first paragraph briefly discusses unisensory development, and then multisensory development (which seems to be beyond the scope of this paper). The second paragraph first discusses older adults (which is again beyond the scope of this paper), how training can prevent decline (which seems again beyond the scope of this paper) and finally multisensory learning in children. Also, throughout the introduction (and the discussion), the authors use numbers to refer to their sources in the middle of sentences, essentially replacing the subject of verbs with numbers - for example “Similarly, in the visual domain (4,5) suggested that infants as young as 6 months old have the ability to retain visual stimuli…” This makes reading the paper quite awkward. The manuscript would benefit from a more clearly focused introduction.

Discussion: The discussion is generally superficial and lacks clear explanations.

• The authors claim that “From the MDS results in Figure 11 it seems that the children may have relied more on fast-outstanding features than on less outstanding details.” What kinds of features and details are the authors referring to here (and how are features and details different)? How do the results support this interpretation? An explanation is needed. In the next sentence the authors refer the reader to the reconstruction in Figure 6 – but Figure 6 represents differences between adults and children’s ratings. They then refer back to the use of petal edges and stems and indicate these features were “recurrent criteria mentioned by children” – however the authors did not report any analysis of this information (they only mention them in the method) – a table of frequent comments would have been helpful, especially with a frequency count. The authors then say that “Changes in feature processing were also observed in (10).” It’s not clear what the authors mean or how those other findings relate to theirs.

• On page 20, the authors describe other research on verbal fluency and state that “This research could also explain why the children group in this experiment did not exhibit changes around a certain age…” How so? The authors should provide this explanation.

• On page 21, the authors claim that “the second trial for adults did not show any kind of improvement in comparison to the first trial” – but no data or statistical analysis is reported

• On page 21, the authors indicate that the children’s reliability may be an issue, but then say “we did observe a greater variability in ratings for children, and their ratings tended to be more similar in absolute terms.” This seems contradictory and needs to be clarified

• On page 21, the authors justify their choice of a similarity rating task – this justification would be more appropriate in the introduction.

• The discussion also includes parts that seem irrelevant / beyond the scope of their paper. For example, on pages 21-22, they discuss hastening multisensory integration through training, which does not seem appropriate here.

6. PLOS authors have the option to publish the peer review history of their article (what does this mean?). If published, this will include your full peer review and any attached files.

Reviewer #1: **Yes: **Francesca Tinelli

Reviewer #2: No

---

## [Author Response · Author response to Decision Letter 0]

25 Nov 2022

In the following responses, each reviewer’s question is prefixed by R[reviewer number]Q[question number], and our answer as R[reviewer number]A[question number]

Reviewer 1

R1Q1: The article is original and well-written. The methodology is well done and the statistical analyses that have been performed are also adequate. the authors have also brought the most critical points into the discussion.

The only weak point of the article in my opinion is the small sample of adult subjects compared to children. Is it not possible to increase it? The results would be even more interesting

R1A1: Thank you for your comment and the positive feedback. As to the number of participants, we have added the following to the “Participants” section of the manuscript to clarify this:

“The sample size for the adult group was set similar to that of studies using this well-documented paradigms with adult participants (17-21) - the sample size for the children group was set to be larger given the potentially larger variability in performance and different attention span following previously published research (e.g., (10,12)).” 

Minor revision:

R1Q2: page 10: The sentence "correlation results....." is not complete

R1A2: The sentence in the manuscript reads “Correlation results only pay attention to the relative differences in the rating profiles, but do not assess absolute differences in the rating behavior.” and seems complete? Perhaps the reviewer can clarify in which way the sentence was incomplete? 

Reviewer 2

R2Q1: No justification was provided for the sample sizes (60 per group for children and 15 per group for adults)

R2A1: We have added the following to the “Participants” section of the manuscript to clarify this:

“The sample size for the adult group was set similar to that of studies using this well-documented paradigms with adult participants (17-21) - the sample size for the children group was set to be larger given the potentially larger variability in performance and different attention span following previously published research (e.g., (10,12)).” 

R2Q2: There is no description of how the data was treated: each child participant provided 45 data points (45 comparisons) and each adult participant provided 90 data points (45 comparisons, twice). The authors only mention that ‘their responses were correlated to each other’ – were each individual ratings for each participant correlated to each individual rating for other participants? Was averaging performed across pairs that were more/less similar? Across each individual’s responses? Also, what type of correlation was calculated?

We apologize for not spelling this out completely and have added an Analysis part that aggregates the information of the result section to make this more transparent.

“Analysis:

The individual ratings of each participant for 45 trials (all trials for the children and only the first repetition for the adults) were analyzed in the following. Ratings were first correlated across all participants via Spearman correlations for all pair-wise comparisons within each modality group to check for inter-rater consistency. We next also correlated the responses for the 45 comparisons across participant groups (but within modality) for all pairwise comparisons to gauge how well response patterns would match. Bootstrapping was done to determine confidence intervals for these comparisons. Since correlations only look at relative changes in responses, we next conducted standard t-tests comparing the adults and children for each of the cells of the 9x9 (symmetric) similarity matrix for each condition, followed by correction for multiple comparisons. 

An analysis of the same-object comparisons was conducted next to determine a ground-truth “accuracy” response as the similarity rating for these trials should be “5”. Two-sample t-tests were run to compare average same-trial ratings across modalities and groups. Linear regressions were then conducted for the children participants to see whether there would be any trend in accuracy as a function of age.

The next set of analyses focused on the similarity matrices and the resulting perceptual space. First, the similarity matrices were compared to those obtained from the actual physical parameters used during stimulus creation to check how well they would match. Next, multidimensional scaling (MDS, e.g., Cooke et al., 2009) was applied to investigate the topologies of the resultant perceptual spaces. Stress values of the MDS and bootstrapped fits of the spaces against each other using Procrustes analysis were performed to determine the degree of topological similarity.”

R2Q3: It is not clear if only first trial or both trials are included in the analysis of adults’ performance. If both trials are included: it is possible that this stabilized participants’ responses and led to the finding that responses were more reliable in adults. If both adult responses were included, it would be important to report the analysis of only the first response, as it may change the results and the authors’ interpretation / conclusion.

R2A3: We fully agree that only the first repetition (trial) should be analyzed and did so in the paper. As now stated more clearly in the analysis part (information that was previously in the Results and Discussion parts), only the first trial was analyzed and reported in the Results section. The second repetition is only used as a measure of consistency to explore the possibility of exposure effects, as mentioned further in the discussion (now with a short quantitative statement).

R2Q4: On page 8 the authors report that age differences were ‘not visible in the data’ – what statistical test was performed to support the statement?

R2A4: Because the results were presented in a correlation matrix, and sorted from youngest to oldest of the children tested, an improvement would have been qualitatively visible as a pattern in the correlation matrix colours (less correlated to more correlated). We have added the following to Page 9:

“In Figure 3, all results are sorted from the youngest to the oldest age to potentially highlight differences - any consistent change across age (such as improvement in consistency for older children) would be visible as a pattern in the correlation matrix colors - this, however, was not observed.”

R2Q5: On most figures the axes are not labeled.

R2A5: Most axes were either similarity matrix item numbers or arbitrary units for which labeling is typically not done. We did, however, omit to include labels for Figure 8, which is now redone with labels - we apologize for the oversight. 

R2Q6: On page 4, figures 3 and 4 occlude text. This also occurs on page 10: Figure 6 occludes text and overlaps with Figure 5, occluding it as well.

R2A6: This may be a problem with the submission system - we have uploaded each figure separately from the manuscript as per the requirements of PLoS One. The captions were also resized to hopefully alleviate the problem.

R2Q7: On page 11-12, the authors perform two separate analyses on the averaged ratings for ‘same’ trials (visual vs. haptic in each age group, and then children vs. adults for each modality). The authors should select one set of comparisons. 

On page 12 it is not clear what data is being used to ‘fit the response of all participants against age in months’.

R2A7: Two analyses were done to first check for differences in same-trial accuracy across adult and children participants and to, second, trace how or whether these responses would change with age for the children participants. Again, this is now more succinctly stated in the “Analysis” section.

R2Q8: Through the results section, the authors seem to begin interpreting their data (for example p 13 for age-related changes, p15 for estimated distances)

R2A8: We have added only intermediate conclusions as we think it would help to guide the reader through the various types of analyses. 

R2Q9: On p. 14 it’s not clear whose data (adults, children, all) is being presented in the analysis of perceptual space. The same problem arises in the first paragraph for multidimensional scaling (page 16).

R2A9: We apologize for the oversight with the MDS stress values - the first paragraph relates to data from the children, whereas the later paragraph on Page 15 now also lists corresponding stress values for the adult data. The remaining paragraphs and figures are, however, labeled and introduced as to which participant group they refer to - perhaps the reviewer could point out which analysis was affected? 

R2Q10: It’s often not clear why specific analyses are being performed, for example, on p 17 the authors say “To compare MDS results further across modalities…” They then indicate they left out participants for the calculation, but did not explain which participants or how they were selected for exclusion.

R2A10: We are sorry for the ambiguous wording - this is purely part of the standard bootstrapping calculation, in which variability of the measure is estimated by repeating the procedure with “left-out” data. This is now reworded as follows:

“This was done by repeating the analysis 1000 times, each time leaving out half of the participants for the calculation of MDS spaces and aggregating the results over repetitions.”

R2Q11: There is no report of how common specific children’s responses were about how they rated objects – though these are addressed in the discussion. It would therefore be important to describe these in the results

R2A11: Given the time-demands placed on us by the school, we were not able to collect these responses in a more structured fashion and hence just noted these down when children made comments about the objects after the experiment had already ended. We have added a comment to this effect to the manuscript on Page 7.

R2Q12: (data availability) The authors report that ‘some restrictions will apply’ but do not describe these restrictions. The authors also indicate that data will be made available upon publication via a Google share link.

R2Q12: Following PLoS One guidelines, data will be made available via an OSF link upon publication - this is what this refers to.

R2Q13: Generally the quality of the English language is appropriate. However, the constant use of numbers in the middle of sentences to refer to specific authors makes reading the manuscript awkward.

R2A13: We agree on the numbering issue - this will be changed during the editing phase, but does (unfortunately) adhere to the PLoS One citation style.

R2Q14: On page 4, the authors should add ‘participants’ after the second 15, and ‘the visual’ should be replaced by ‘the visual task’ in the sentence “A total of 15 participants performed the haptic task and 15 (participants) performed the visual (task).”

R2A14: Thank you - we have made the requested changes.

R2Q15: Further to the above, the authors should address the following:

Introduction: The introduction is very short (2 pages / 3 paragraphs), quite superficial, and lacks focus. The first paragraph briefly discusses unisensory development, and then multisensory development (which seems to be beyond the scope of this paper). The second paragraph first discusses older adults (which is again beyond the scope of this paper), how training can prevent decline (which seems again beyond the scope of this paper) and finally multisensory learning in children. Also, throughout the introduction (and the discussion), the authors use numbers to refer to their sources in the middle of sentences, essentially replacing the subject of verbs with numbers - for example “Similarly, in the visual domain (4,5) suggested that infants as young as 6 months old have the ability to retain visual stimuli…” This makes reading the paper quite awkward. The manuscript would benefit from a more clearly focused introduction.

R2A15: Please see above for the issue with numbering references. We had tried to focus the introduction (see also R2Q20 below), but there is - at present - not enough related work in the area of developmental research in this context, such that we opted to include the work on multisensory integration. We think, however, that the streamlined Analysis, Results, and Discussion section in our revision make for a better reading experience overall. 

R2Q16: Discussion: The discussion is generally superficial and lacks clear explanations.

• The authors claim that “From the MDS results in Figure 11 it seems that the children may have relied more on fast-outstanding features than on less outstanding details.” What kinds of features and details are the authors referring to here (and how are features and details different)? How do the results support this interpretation? An explanation is needed. In the next sentence the authors refer the reader to the reconstruction in Figure 6 – but Figure 6 represents differences between adults and children’s ratings. They then refer back to the use of petal edges and stems and indicate these features were “recurrent criteria mentioned by children” – however the authors did not report any analysis of this information (they only mention them in the method) – a table of frequent comments would have been helpful, especially with a frequency count. The authors then say that “Changes in feature processing were also observed in (10).” It’s not clear what the authors mean or how those other findings relate to theirs.

R2A16: We apologize for being unclear about this. Our qualitative interpretation of these findings relate to the features that were mentioned by the children spontaneously during the experiment - unfortunately, we were not able to conduct structured post-experiment debriefings due to time constraints. This is now clarified on Page 7:

“Given the time constraints placed on this experiment, it was not possible to conduct a more detailed, structured post-experiment debriefing unfortunately.”

and in the discussion by adding that we conducted only a qualitative assessment and concluding as follows:

“A more in-depth analysis of the introspective features of children versus adults in this task, however, was not possible in the context of the present study, given the time constraints in running the experiment in the school and hence is left for future work.”

R2Q17: On page 20, the authors describe other research on verbal fluency and state that “This research could also explain why the children group in this experiment did not exhibit changes around a certain age…” How so? The authors should provide this explanation.

R2A17: We have added the following to the Discussion to clarify our thinking:

“ - this is because verbal fluency development would have an impact for a familiar object task, whereas the objects in our experiment were unfamiliar making it potentially more difficult to attach clear semantic associations.”

R2Q18: On page 21, the authors claim that “the second trial for adults did not show any kind of improvement in comparison to the first trial” – but no data or statistical analysis is reported

R2A18: This is now clarified and we have updated this to read as follows: 

“One potential limitation of our experiment is that we were only able to gather results from a single trial per comparison due to time restrictions and concentration of the younger children. In this context, it is interesting to note that the second trial for adults did not show differences in comparison to the first trial (correlation of r=.87 of first repetition to second repetition in the ratings). Similarly, the children’s same trial ratings were consistent at similar levels to those of the adults, showing that the ratings per se were not unreliable.”

R2Q19: On page 21, the authors indicate that the children’s reliability may be an issue, but then say “we did observe a greater variability in ratings for children, and their ratings tended to be more similar in absolute terms.” This seems contradictory and needs to be clarified

R2A19: We apologize for this imprecisely stated sentence - the second part is not relevant and was now deleted. We did, however, observe greater variability in responses across children compared to adults (see Results section).

R2Q20: On page 21, the authors justify their choice of a similarity rating task – this justification would be more appropriate in the introduction.

R2A20: We have tried to fit this section into the introduction, but have reverted it back to the discussion in our revision. The introduction already contains a justification of why we chose to use this task based on prior work in adults on exactly this paradigm. The paragraph the reviewer refers to, however, relates more to potential issues with similarity ratings and their reliability (see previous answer), and we therefore felt that it would still be better suited for the discussion section. 

R2Q21: The discussion also includes parts that seem irrelevant / beyond the scope of their paper. For example, on pages 21-22, they discuss hastening multisensory integration through training, which does not seem appropriate here.

R2A21: We agree and have now deleted this section.

---

## [Decision Letter · Decision Letter 1]

29 Dec 2022

PONE-D-22-15932R1Visuo-haptic processing of unfamiliar shapes: comparing children and adultsPLOS ONE

Dear Dr. Wallraven,

Thank you for submitting your manuscript to PLOS ONE. After careful consideration, we feel that it has merit but does not fully meet PLOS ONE’s publication criteria as it currently stands. Therefore, we invite you to submit a revised version of the manuscript that addresses the points raised during the review process.

We look forward to receiving your revised manuscript.

Kind regards,

Mariella Pazzaglia

Academic Editor

PLOS ONE

Reviewers' comments:

Reviewer's Responses to Questions

**Comments to the Author**

1. If the authors have adequately addressed your comments raised in a previous round of review and you feel that this manuscript is now acceptable for publication, you may indicate that here to bypass the “Comments to the Author” section, enter your conflict of interest statement in the “Confidential to Editor” section, and submit your "Accept" recommendation.

Reviewer #1: (No Response)

Reviewer #2: All comments have been addressed

2. Is the manuscript technically sound, and do the data support the conclusions?

Reviewer #1: Partly

Reviewer #2: Yes

3. Has the statistical analysis been performed appropriately and rigorously? 

Reviewer #1: Yes

Reviewer #2: Yes

4. Have the authors made all data underlying the findings in their manuscript fully available?

Reviewer #1: Yes

Reviewer #2: Yes

5. Is the manuscript presented in an intelligible fashion and written in standard English?

Reviewer #1: Yes

Reviewer #2: Yes

6. Review Comments to the Author

Reviewer #1: The article is very interesting especially because it concerns what happens in the developmental age. The authors have made good statistical analyses that are also well represented graphically.

However, I have three major concerns:

1) there is a lack of description of the visual experiment (it is taken for granted that one understands what is being done but I cannot find any explanation and instead believe it would be useful to explain what is being done and asked of the subject)

2) I am very amazed that the haptic modality seems to be more helpful than the visual modality in children and so I wonder how they are made to see these shapes. the author's did not report any infomration about vision of the enrolled children. can you specifiy if they have all a normal vision, if the experimetn was done with glasses if they need ....

3) I wonder if comparing a population of children from one country with that of adults from a country with very different habits can really be correct.

Minor revision:

In the introduction you refer to the age at which visual acuity and contrast sensitivity develops, but you don't specify whether you are talking about months or years. since in the previous paragraph the reference was to months I think it should be better specified what the authors mean here

Translated with www.DeepL.com/Translator (free version)

Reviewer #2: I am quite happy with the revisions - the authors did a thorough job. I wish to thank the authors for addressing my concerns!

7. PLOS authors have the option to publish the peer review history of their article (what does this mean?). If published, this will include your full peer review and any attached files.

Reviewer #1: No

Reviewer #2: No

---

## [Decision Letter · Decision Letter 2]

21 Feb 2023

PONE-D-22-15932R2Visuo-haptic processing of unfamiliar shapes: comparing children and adultsPLOS ONE

Dear Dr. Christian Wallraven,

Thank you for submitting your manuscript to PLOS ONE. After careful consideration, we feel that it has merit but does not fully meet PLOS ONE’s publication criteria as it currently stands. Therefore, we invite you to submit a revised version of the manuscript that addresses the points raised during the review process.

We look forward to receiving your revised manuscript.

Kind regards,

Mariella Pazzaglia

Academic Editor

PLOS ONE

Additional Editor Comments (if provided):

Thank you for your revised version of this manuscript. As you can see, one reviewer have raised significant concern about this study. Please address these concerns raised by reviewer

Reviewers' comments:

Reviewer's Responses to Questions

**Comments to the Author**

1. If the authors have adequately addressed your comments raised in a previous round of review and you feel that this manuscript is now acceptable for publication, you may indicate that here to bypass the “Comments to the Author” section, enter your conflict of interest statement in the “Confidential to Editor” section, and submit your "Accept" recommendation.

Reviewer #3: (No Response)

2. Is the manuscript technically sound, and do the data support the conclusions?

Reviewer #3: Partly

3. Has the statistical analysis been performed appropriately and rigorously? 

Reviewer #3: Yes

4. Have the authors made all data underlying the findings in their manuscript fully available?

Reviewer #3: Yes

5. Is the manuscript presented in an intelligible fashion and written in standard English?

Reviewer #3: Yes

6. Review Comments to the Author

Reviewer #3: My comments are uploaded as an attachment and copied below.

This is an interesting paper on a timely topic, looking at the similarity in how adults and children rate objects they explore via touch versus via looking. Rather than consider integration between the senses, they consider a given each sensory modality individually. They focus on children 6-11 years of age as they highlight that changes in the processing of vision and touch have been established around 8 to 9 years of age. Importantly, they use ratings for unfamiliar objects where they parametrically vary stimulus dimensions for visual and 3D printed haptic objects, a total of 9 unique objects presented in all possible pairings, for 45 trials total. A 5 point Likert scale was used to obtain ratings.

While I thought the approach of the paper was strong, the methods and analyses were not always straight forward.

MAJOR COMMENTS

1.The authors had 30 adult participants, 15 in the haptic group and 15 in the vision group. Each adult completed 2 repetitions of the experiment and used a numerical scale to give their ratings of the similarity between 1 given pair of objects. Unlike the adult data, 126 children participated, with a final of 30 children in the haptic group and 30 children in the vision group. Each child ran only 1 repetition of the experiment and provided verbal ratings. How could these discrepancies contribute to the results of the analysis? How do verbal vs numerical ratings compare,? Why was this aspect of the study changed? Also, how did the authors determine that they needed to double the amount of data collected in children to account for the additional variance in younger cohorts?

The author’s state that the children’s age ranges from 6-11 (although figure 8 also seems to go to 125 months, or age 10.5 years), yet no range is provided for the adults, only the mean of 24.8 years is specified. It would be good to know the spread in age for both groups since it is stated that results are sorted by age. A broader spread in the age of child participants might contribute to more variability. Also, what was the exclusion criterion for not including data?

2. While IRB approval is mentioned for the adult study in Korea, no IRB approval is mentioned for studies run in Germany in the school setting. Was an IRB not obtained for this work?

3. It is great that the abstract shapes are created by parametrically varying certain stimulus dimensions. What those dimensions are is not clear in Figure 1. Some background as to how the parameterization was done and why should be provided.

4. The methods state that children were given 6-8 seconds to explore the first object before being given the second object and that adults were given 6 seconds. Was this sufficient time? The authors state that children found the task easy, but is there some measure of mean reaction time in their making their judgments? One would imagine children might need more time to explore the shapes, especially via touch, than adults.

5. It is not clear what the correlation matrices in Fig 3 and 4 are plotting. The x and y axes are not labelled. The text mentions that results are sorted by age, but it is not clear how this information is being represented. Also, Figure 3 and 4 should have a part (a) and (b), as mentioned in the text, but the figure is missing those labels.

6. What does the color coding in Figure 6 indicate? If it is level of significance, there should be clear indication of what each color means in terms of significance or size of the effect.

MINOR COMMENTS:

Highlighted in yellow are places where the authors use a number for a citation in an odd way or places where an article needs to be changed, with the changed word added.

Page 7: Change “to haptically fully explore (Haptic group)” � “to fully explore via touching (Haptic group) and then to fully explore via looking (Visual group)”

Page 15: The stress values obtained for judging the goodness-of-fit of a two-dimensional scaling solution were s = 0.04 for the haptic condition and s = 0.04 for the visual condition which is determined as a “good” fit. (17) (23). – extra period should be eliminated

In Figures, differences that are significant should have a clear label and the figure legend should indicate the p value. For example, Figure 13, it is not clear what the “+” indicates

Page 15: “The children’s performance was as good as the adults’ and there was no advantage OF age in the categorization task.

Page 19: “To summarize, in this experiment we looked into the developmental trajectory of haptic and visual processing FOR unfamiliar objects as opposed to the integration of both senses in elementary school children.

7. PLOS authors have the option to publish the peer review history of their article (what does this mean?). If published, this will include your full peer review and any attached files.

Reviewer #3: No

---

## [Author Response · Author response to Decision Letter 2]

5 May 2023

In the following responses, each reviewer’s question is prefixed by R[reviewer number]Q[question number], and our answer as R[reviewer number]A[question number]

Reviewer 3

R3Q1) The authors had 30 adult participants, 15 in the haptic group and 15 in the vision group. Each adult completed 2 repetitions of the experiment and used a numerical scale to give their ratings of the similarity between 1 given pair of objects. Unlike the adult data, 126 children participated, with a final of 30 children in the haptic group and 30 children in the vision group. Each child ran only 1 repetition of the experiment and provided verbal ratings. How could these discrepancies contribute to the results of the analysis? How do verbal vs numerical ratings compare? Why was this aspect of the study changed? Also, how did the authors determine that they needed to double the amount of data collected in children to account for the additional variance in younger cohorts?

R3A1) While the adults completed two repetitions, only the first repetition was used in the main analysis and comparisons to children’s performance (see Page 8, Setup “Adults Similarity Task”, but cf. Discussion Pages 18/19 for a brief discussion of the second repetition). Sample sizes were determined via the cited publications (see Page 5: “The sample size for the adult group was set similar to that of studies using this well-documented paradigms with adult participants (17-21) - the sample size for the children group was set to be larger given the potentially larger variability in performance and different attention span following previously published research (e.g., (10,12)).”). There were 60 children in each group. 

The verbal responses for the children were also accompanied by a number rating and were only used as a guideline for them to make sure they understood the ratings and the meanings assigned to each number. Many children did, actually, provide numerical ratings in their answers. 

R3Q2) The author’s state that the children’s age ranges from 6-11 (although figure 8 also seems to go to 125 months, or age 10.5 years), yet no range is provided for the adults, only the mean of 24.8 years is specified. It would be good to know the spread in age for both groups since it is stated that results are sorted by age. A broader spread in the age of child participants might contribute to more variability. Also, what was the exclusion criterion for not including data?

R3A2) There was a typo for the average range of adults - it is 22.8 years - we are sorry for this oversight. The age range for the adults is now included as well; given this and the two standard deviations of the two groups (1.2 for children versus 2.2 for adults), we note that standard deviation for children is actually smaller than that for adults with age range being roughly similar. 

Exclusion criteria were only applied for data of children and - as stated in the manuscript - related to children not being able to complete the full experiment (not all trials could be run), or failing to understand the instructions within the allotted time slots (participants did not look at the object in the visual condition despite being instructed several times to do so). This only affected 5 participants in total.

R3Q3) While IRB approval is mentioned for the adult study in Korea, no IRB approval is mentioned for studies run in Germany in the school setting. Was an IRB not obtained for this work?

R3A3) IRB approval was obtained for both countries, and the German ethics approval was referenced in the manuscript - please see page 4, “Participants” section “Ethics proposal number: Rau_2018_1025_138”. 

R3Q4) It is great that the abstract shapes are created by parametrically varying certain stimulus dimensions. What those dimensions are is not clear in Figure 1. Some background as to how the parameterization was done and why should be provided.

R3A4) The dimensions manipulated in creating the shapes were based on parameters (n1, m1, and m2) from the Superformula equation and followed the approach in publication (21). The “dimensions” themselves are simply three of the parameters that can be varied in the full set of equations - these all result in global shape changes, such that it is hard to give an intuition about their effects on the shape. We have added the following to the text on page 5: 

“We note that the parameters introduce global shape changes that are hard to summarize easily - for more discussion on these stimuli, see (21).”

R3Q5) The methods state that children were given 6-8 seconds to explore the first object before being given the second object and that adults were given 6 seconds. Was this sufficient time? The authors state that children found the task easy, but is there some measure of mean reaction time in their making their judgments? One would imagine children might need more time to explore the shapes, especially via touch, than adults.

R3A5) The duration of the exploration is based on previous haptic experiments of a similar nature (eg, 21) and leaves ample time for exploration. We conducted pilot tests with a few young children and found that this, also, is sufficient time. The setting of the experiment did not allow us to record “proper” reaction times, but we did observe anecdotally that in the vast majority of trials, children answered within 2-4s.

R3Q6) It is not clear what the correlation matrices in Fig 3 and 4 are plotting. The x and y axes are not labelled. The text mentions that results are sorted by age, but it is not clear how this information is being represented. Also, Figure 3 and 4 should have a part (a) and (b), as mentioned in the text, but the figure is missing those labels.

R3A6) The figure labels have been updated and we have expanded the explanation now on page 9 to read as follows:

“Consistency across participants was the first step in analysis to assure the reliability of the results. To measure participants’ rating consistency, their similarity rating responses for all object pairs were correlated to each other resulting in the correlation matrices shown in (Fig. 3). By averaging their lower diagonal, response consistency yielded an average correlation of r = .56 for haptic (Fig. 3a), and r = .55 for visual exploration (Fig. 3b). In Figure 3, all results are sorted from the youngest to the oldest age to potentially highlight differences - any consistent change across age (such as improvement in consistency for older children) would be visible as a pattern in the correlation matrix colors - this, however, was not observed.”

R3Q7) What does the color coding in Figure 6 indicate? If it is level of significance, there should be clear indication of what each color means in terms of significance or size of the effect.

R3A7) The color is proportional to the actual difference in the similarity ratings for each object pair comparison. Colors are only used for cells surviving the FDR correction. This has now been clarified in the figure caption and we have included a color bar scale with each figure. 

R3Q8) Page 7: Change “to haptically fully explore (Haptic group)” -> “to fully explore via touching (Haptic group) and then to fully explore via looking (Visual group)”

R3A8) This was changed to “to fully explore via touching (Haptic group) or to fully explore via looking (Visual group)” - since each group only ever experience the objects in their corresponding modality.

R3Q9) Page 15: The stress values obtained for judging the goodness-of-fit of a two-dimensional scaling solution were s = 0.04 for the haptic condition and s = 0.04 for the visual condition which is determined as a “good” fit. (17) (23). – extra period should be eliminated 

R3A9) Thank you. This was corrected.

R3Q10) In Figures, differences that are significant should have a clear label and the figure legend should indicate the p value. For example, Figure 13, it is not clear what the “+” indicates

R3A10) In Figure 13 the “+” sign is an indicator of an outlier according to the standard box-whisker plotting style - we have added this explanation to the caption.

R3Q11) Page 15: “The children’s performance was as good as the adults’ and there was no advantage OF age in the categorization task.

Page 19: “To summarize, in this experiment we looked into the developmental trajectory of haptic and visual processing FOR unfamiliar objects as opposed to the integration of both senses in elementary school children.

R3A11) Thank you for these suggestions. We have updated this in the revision.

---

## [Decision Letter · Decision Letter 3]

26 May 2023

Visuo-haptic processing of unfamiliar shapes: comparing children and adults

PONE-D-22-15932R3

Dear Dr. Christian Wallraven,

We’re pleased to inform you that your manuscript has been judged scientifically suitable for publication and will be formally accepted for publication once it meets all outstanding technical requirements.

Kind regards,

Mariella Pazzaglia

Academic Editor

PLOS ONE

Additional Editor Comments (optional):

Reviewers' comments:

Reviewer's Responses to Questions

**Comments to the Author**

1. If the authors have adequately addressed your comments raised in a previous round of review and you feel that this manuscript is now acceptable for publication, you may indicate that here to bypass the “Comments to the Author” section, enter your conflict of interest statement in the “Confidential to Editor” section, and submit your "Accept" recommendation.

Reviewer #3: All comments have been addressed

2. Is the manuscript technically sound, and do the data support the conclusions?

Reviewer #3: Yes

3. Has the statistical analysis been performed appropriately and rigorously? 

Reviewer #3: Yes

4. Have the authors made all data underlying the findings in their manuscript fully available?

Reviewer #3: Yes

5. Is the manuscript presented in an intelligible fashion and written in standard English?

Reviewer #3: Yes

6. Review Comments to the Author

Reviewer #3: The authors have addressed all of my major and minor concerns. Issues I had with figures and data have been clarified and small grammatical details have been corrected. The manuscript is improved and clearer.

7. PLOS authors have the option to publish the peer review history of their article (what does this mean?). If published, this will include your full peer review and any attached files.

Reviewer #3: No

---

## [Editor Report · Acceptance letter]

19 Oct 2023

PONE-D-22-15932R3 

Visuo-haptic processing of unfamiliar shapes: comparing children and adults 

Dear Dr. Wallraven:

I'm pleased to inform you that your manuscript has been deemed suitable for publication in PLOS ONE. Congratulations! Your manuscript is now with our production department. 

Kind regards, 

on behalf of

Dr. Mariella Pazzaglia 

Academic Editor

PLOS ONE